# A Modular Abstraction for Integrating Domain Rules into Deep Learning Models

## Abstract

Domain-specific knowledge can often be expressed as suggestive rules defined over subgroups of data. Such rules, when encoded as hard constraints, are often not directly compatible with deep learning frameworks that train neural networks over batches of data. Also, domain-experts often use heuristics that should not be encoded as logical rules. In this work, we propose a framework to capture domain-experts' knowledge as domain-specific rules over subgroups of data, and to leverage such rules in training deep learning models using the modular components of regularization, data augmentation, and parameter optimization. This translation of domain knowledge into custom primitives that can be augmented to existing state-of-the-art deep learning models improves the ability of domain experts to interpret and express model behavior, intervene through changes in the modeling specifications, and improve the overall performance of the model as compared to existing frameworks that incorporate deterministic declarative predicates. On one synthetic and three real-world tasks, we show that our method allows iterative refinement and is demonstrably more accurate.

## 1 Introduction

Black box deep learning models trained only on observed historical data generally contradict rules derived from domain knowledge (e.g., obtained through experience or meticulous validation using intervention based studies like randomized control trials). Domain experts, especially in safety-critical domains, are particularly skeptical of such models, which limits their widespread adoption and deployment (O'Mahony et al., 2005; Sinha et al., 2017; Xin et al., 2017). Incorporating domain knowledge in deep learning models not only gains the trust of domain experts, but can also drastically improve the performance of the model, especially in unseen circumstances (Towle & Quinn, 2000). Hence, we propose a framework for capturing and integrating domain knowledge in the form of loosely defined rules, which we then translate into regularization, data augmentation, and parameter optimization modules to train and evaluate deep learning models.

In domains like health, socio-economic inference, and content moderation, a fundamental roadblock for developing deep learning systems is that the predictions diverge from domain knowledge when deployed in the real world and fail to incorporate domain-specific structure in counterfactual data distributions. However, domain knowledge is often intuitive or experiential, and therefore hard to capture. For example, consider the medication recommendation task where doctors use patient symptoms as input and recommend a medication as output. Here, although medical ontologies provide guidance on how medications should be prescribed conditional on the patient symptoms, they are not considered to be exhaustive. In fact, doctors use several unstructured sources of priors that are not effectively captured in medical ontologies directly, yet influence their prescriptions. Hence, it is important to use the medical ontologies as priors when they are available, and rely on data-driven patterns when they are not. However, the dichotomy is not trivial as within a single patient's history, some symptoms may be covered by the ontologies, whereas others are not - leading to interactions between these sub-spaces that require additional probabilistic modeling.

Using our framework, domain experts are able to define domain knowledge as suggestive rules and effectively modularize those rules in terms of loss functions, data distributions or model hypothesis space. For example, if a doctor has certain preferred medications for patients presenting with dermatological conditions on the

hand, they can specify this preference using a mapping rule. Within the properties of this rule they can then specify that it should hold over data augmentations performed on the fly using a custom loss function for concept-based regularization loss. Each of the modules for data augmentation and regularization is readily available to the doctor with parameters they can choose and optimize as and when their data distributions and priors continue to evolve. Thus, our framework provides domain experts visibility into the entire machine learning pipeline with the ability to intervene in each of the modules of the pipeline through programmatic specifications. Compared to prior work that primarily requires specifying priors over models using a family of labeling functions, our approach reduces the need for domain experts to explicitly construct labeling functions, a task that can be challenging in practice. While we have not conducted a formal user study, we expect that this modular, rule-based specification can be more approachable for domain experts than approaches centered on labeling functions.

**Our contributions include**: **(1)** A framework based on data augmentation and regularization which we show provides statistical guarantees based on probabilistic robust learning Robey et al. (2022). **(2)** When evaluated on synthetic discontinuous and non-differential constraints, our framework performs equally well as Bayesian modeling techniques (Tavares et al., 2019) **(3)** We outperform the G-BERT baseline on MIMIC-III medication recommendation task by 19.1% and a corresponding manually augmented and regularized baseline by 5.3% in area under the precision-recall graph. **(4)** We also apply our method to the Car Racing Box2D reinforcement learning task, and show that it consistently outperforms Deep Q learning baseline models (DQN) that do not incorporate map outlines, and reduces the number of training episodes to achieve a reward of 350 points by 50% ($100 \rightarrow 50$). **(5)** When applied to the task of toxicity detection, our framework shows an improvement of 23.4% over Perspective-BERT transformer model.

## 2 Related Work

Prior work on incorporating domain knowledge in deep learning systems spans the research areas of robustness, recommendation systems, human-computer interaction, constraint learning, and neuro-symbolic computing. Robustness research in deep learning is mainly focused on the concept of adversarial robustness, which while important, is orthogonal in scope for the use cases domain experts care about. Thus, domain specified notion of robustness, like in the field of computer vision, robust models over concept-based perturbations (Xie et al., 2019) and in natural language processing (Hsieh et al., 2019), robustness over word substitutions with synonyms are desired (Qin et al., 2019).

In the field of recommendation models, there has been a lot of interest in making ML systems avoid extremely undesired outcomes (e.g. horror films to children) (Wang & Caverlee, 2019; Xin et al., 2017). Another example of such prohibitive constraints is large language models where guardrails are enforced to ensure that the models do not generate ethically unsound text. Such guardrails are another example of domain knowledge that can be incorporated in deep learning models. Further, in the biomedical natural language processing field, recent work has trained domain-specific ML models starting from pre-training, and fine-tuning (Lee et al., 2020) to perform question-answering on medical factoid questions (Nentidis et al., 2025). We use such customized domain-specific baselines (e.g. G-BERT (Shang et al., 2019) for medical purposes) as state-of-the-art models to apply our methodology.

There are other methods in the literature that allow rule specification in the form of logic programming. The rules are often specified as grammars, and probabilistic rules are represented using Stochastic Definite Clause Grammars (SDCGs) (Have, 2009). These works enable rule specification as predicates in Prolog and allow inference through tree-based backtracking. The most relevant work (Winters et al., 2022) extends this approach by allowing the integration of neural rules using Neural Definite Clause Grammar (NDCG). However, during goal resolution, it can be very expensive if the branching factor for the Selective Linear Definite clause resolution tree becomes large.

Another line of work looks into the development of hybrid human-computer systems to aid the domain experts in interpreting the machine learning model's predictions (Gennatas et al., 2020; Villena-Román et al., 2011) and guiding the underlying deep learning model through interactive feedback (Cai et al., 2019) and inductive logic (Wilcox & Hripcsak, 2003) that aligns the model's predictions to domain knowledge, as done by (Morik et al., 1999) in the medical domain. Other approaches have looked into mapping human interpretable rules

with ML models to understand the inner workings of a black-box machine learning model. (Doshi-Velez & Kim, 2017) defines the "task related latent dimensions of interpretability" where, we care about the hypothesis of local interpretability(Ribeiro et al., 2016), with incomplete coverage of domain expertise (Zhang et al., 2018b), which is closely related to our approach of adding interpretable domain-specific rules to deep-learning models. Additionally, there is work Hu et al. (2016) that looks into iteratively distilling knowledge in the form of logical rules into the neural network framework.

Domain knowledge often imposes hard constraints on a model in the sense that the constraints exclude parts of the sample space altogether rather than only making them less likely. This means the target distribution may have isolated modes and sharp derivatives, causing poor sampler exploration and instability in gradient-based methods. In these challenging cases, another line of work on incorporating constraints in deep learning systems looks into approximate inferences. For instance, simulated annealing (Kirkpatrick et al., 1983) can be used to bring the posterior closer to a uniform density, potentially connecting isolated modes. Parallel tempering samples from a collection of exact and approximate models (*replicas at increasing temperatures*) in parallel (Swendsen & Wang, 1986; Geyer, 1991; Earl & Deem, 2005; Altekar et al., 2004; Syed et al., 2019). Predicate Exchange (Tavares et al., 2019) applies parallel tempering to cases where sets are defined by hard predicates expressing arithmetic constraints.

Also, there have been many works in neuro-symbolic (NeSy) computing (d'Avila Garcez et al., 2019; **?**; Giunchiglia et al., 2022; Manhaeve et al., 2018; Li et al., 2023; 2020) that can incorporate logical rules in deep learning models. While some of these methods can encode relatively expressive formalisms such as first-order logic (Badreddine et al., 2022) or logic programming, their applicability is often constrained by the requirement that domain knowledge be formalized in a symbolic logic form. In many real-world settings, especially where domain expertise is heuristic, probabilistic, or loosely specified, expressing such knowledge entirely in logical form can be difficult or impractical. More recent work like (Magnini et al., 2022) allow architecture-agnostic interventions by weighting the gradient updates, while (De Smet et al., 2025) scale to symbolic knowledge in sequential decision-making tasks. We use (Winters et al., 2022) as a representative baseline method from this body of work as its performance on challenging neural symbolic learning tasks is comparable to state-of-the-art methods. We refer to (Ciatto et al., 2024) for a systematic review of symbolic knowledge injection methods.

One work that is conceptually similar to our approach is Asai & Hajishirzi (2020), which uses logic-guided data augmentation and regularization to enforce consistency with predefined rules in question answering. However, Asai & Hajishirzi (2020) operates on deterministic logical constraints in a single NLP domain, whereas our Domain Faithful Deep Learning framework generalizes to probabilistic and heuristic rules, supports modular specification over loss functions, data distributions, and model spaces, and is applicable across diverse domains such as healthcare, reinforcement learning, and content moderation.

## 3 Learning Modular Domain Specific Models

To develop domain-specific models—those that encode domain expertise and rules into their predictions—it is essential to understand how to effectively specify such rules in a way that complements deep neural network (DNN) learning paradigms. In this section, we formally define the framework employed to integrate domain expert rules with DNNs. We describe the implementation methodology and delineate the problem structure, focusing on how expert knowledge is systematically incorporated into the learning process. This foundation provides the basis for comprehending how expert input is leveraged within the broader context of contemporary machine learning models.

In a supervised learning setting, consider an input space $\mathcal{X}$, an output space $\mathcal{Y}$, and an annotated dataset $D = \{(x_i, y_i)\}_{i=1}^{N}$, where $N$ represents the number of input-output pairs, with $x_i \in \mathcal{X}$ and $y_i \in \mathcal{Y}$. The combination of the dataset $D$, along with the input and output spaces $\mathcal{X}$ and $\mathcal{Y}$, constitutes the `DATA-SPACE`. In alignment with common deep learning methodologies, the goal is to train a deep neural network (DNN) model $\mathcal{M} : \mathcal{X} \to \mathcal{Y}$ such that $\mathcal{M}(x_i) = y_i \quad \forall\ (x_i, y_i) \in D$. The typical approach involves minimizing an application-specific loss function $\mathcal{L}_{app}$ defined over the dataset $D$ as follows:

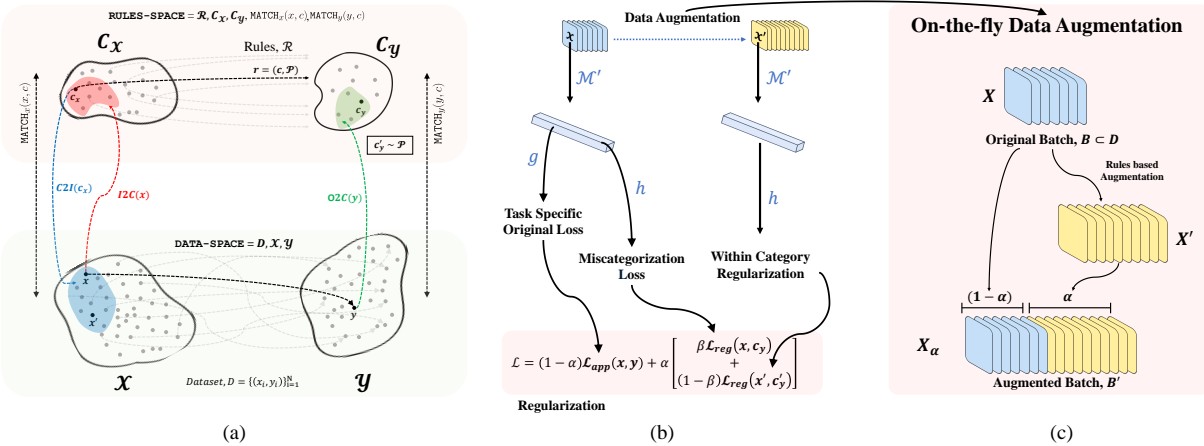

Figure 1: Workflow of incorporating domain-specific rules in deep learning models. (a) Visualization of the interactions in the `RULES-SPACE` and the `DATA-SPACE` as used in Algorithm 1. (b) per batch regularization losses and their combination. (c) On-the-fly batch augmentation.

$$\mathcal{L}_{app}(D) = \mathop{\mathbb{E}}_{(x_i, y_i) \in D} \texttt{Error}(\mathcal{M}(x_i), y_i)$$

Here, `Error` denotes a metric used to quantify the deviation between the model's predictions $\mathcal{M}(x_i)$ and the true outputs $y_i$. For example, in classification tasks, $\mathcal{L}_{app}$ is often instantiated as the cross-entropy loss, while for regression tasks, it typically corresponds to the mean squared error loss. This loss can be computed for a mini-batch $B \subseteq D$ and is minimized using optimization algorithms such as stochastic gradient descent (SGD) or Adam (Kingma & Ba, 2017).

## 3.1 Domain Rules

To construct domain-specific models, it is necessary not only to train the model $\mathcal{M}$ on the dataset $D$ as previously described but also to ensure that the model adheres to a set of domain-specific expert rules when making predictions on $D$. Drawing inspiration from the framework proposed by (Balashankar et al., 2021a), we introduce a structure that enables the integration of these expert rules in a way that is (i) easy to specify, (ii) reflective of how rules are articulated in real-world contexts, and (iii) seamlessly compatible with current deep neural network (DNN) learning pipelines.

However, unlike the framework introduced by (Balashankar et al., 2021a), we propose a more general approach that allows for the specification of uncertainty in domain expert rules. In contemporary data-driven machine learning, it is often challenging to obtain large-scale datasets of manually annotated domain expert rules. Consequently, rules are frequently aggregated over substantial collections, which introduces uncertainty in individual rule specifications. To accelerate this process, many deep neural network (DNN) models are employed in upstream tasks. Their logits or softmax values—particularly in the case of calibrated neural networks—can serve as uncertainty estimates for rule mappings

We now formally define this structure and introduce the relevant notations. Let $\mathcal{C}_x$ represent the set of high-level abstract concept classes over the input space $\mathcal{X}$, and similarly, let $\mathcal{C}_y$ denote the concept classes over the output space $\mathcal{Y}$. An **exact** domain rule is defined as a tuple $(c_x, c_y)$, where a concept class $c_x \in \mathcal{C}_x$ is considered to be *related* to a concept class $c_y \in \mathcal{C}_y$. This rule can be interpreted as a mapping from $c_x$ to $c_y$. In contrast, a **soft** rule refers to a relationship between $c_x$ and $c_y$ with some degree of uncertainty. A common method for representing these uncertainties is to conceptualize the rules as a joint distribution on $c_x$ and $c_y$, denoted as $P(c_x, c_y)$. However, despite the presence of uncertainty, soft rules are rarely provided as explicit joint distributions. In most cases, they are expressed in a conditional form, conditioned on the input class $c_x$.

Formally, a rule, denoted as $r$, is defined for an abstract input concept class $c \in C_x$ as a probability distribution over all the classes in $C_y$. This distribution is expressed as $\mathcal{P} = P(c_y \mid c_x = c)$, allowing the rule to be represented as a tuple $r = (c, \mathcal{P})$. Defining a soft rule as a conditional distribution is particularly valuable for practitioners, as it aligns with common data collection practices which are used to infer rules.

Also note that there can be multiple rules on the same class, $c$. A *soft* rule $r$ can be reduced to an *exact* rule by setting $\mathcal{P}$ as a point mass distribution concentrated on a single class in $C_y$. Henceforth, we will refer to **soft** rules simply as rules, and the set of all such rules $r$'s will be denoted by $\mathcal{R}$.

In addition to specifying the rules, it is necessary to define the mapping between instances in the input space $\mathcal{X}$ and their associated concept classes in $\mathcal{C}_x$. This mapping is governed by a logic that we abstract under the function $\texttt{MATCH}_x(x, c_x)$, which is provided by domain experts. The function $\texttt{MATCH}_x(x, c_x)$ holds true if the input instance $x$ belongs to the concept class $c_x$, as defined by expert logic. Similarly, we can define $\texttt{MATCH}_y(y, c_y)$ for the output side.

We define several important set generation functions that are essential for our implementation. The function $I2C$ maps an input data point to the set of concepts it is associated with, while $O2C$ maps an output data point to the corresponding set of concepts. The function $C2I$ maps a concept to the set of input data points that subscribe to it, and $C2O$ performs a similar mapping for output data points. Finally, $C2R$ maps a concept to the set of rules that are associated with it.

$$I2C(x) = \{c : \texttt{MATCH}_x(x, c) \quad \text{and} \quad c \in C_x\}$$

$$O2C(y) = \{c : \texttt{MATCH}_y(y, c) \quad \text{and} \quad c \in C_y\}$$

$$C2I(c_x) = \{x : \texttt{MATCH}_x(x, c_x) \quad \text{and} \quad x \in \mathcal{X}\}$$

$$C2O(c_y) = \{y : \texttt{MATCH}_y(y, c_y) \quad \text{and} \quad y \in \mathcal{Y}\}$$

$$C2R(c_x) = \{r = (c, \mathcal{P}) : \ c = c_x \quad \forall r \in \mathcal{R}\}$$

The matching function: $f \colon (\texttt{MATCH}_x, \texttt{MATCH}_y)$, along with the concept classes $\mathcal{C}_x$, $\mathcal{C}_y$, and the domain rules $\mathcal{R}$, constitutes the $\texttt{RULEs-SPACE}$. This structure allows for a clear separation between the $\texttt{DATA-SPACE}$ and the $\texttt{RULEs-SPACE}$ (see Figure 1(a)), mirroring real-world scenarios where rules are typically independent of the data they regulate. Another advantage of this separation is that domain experts can modify the rules and logics without altering the original dataset $D$. The enforcement of these rules in the learning process happens through data augmentation and regularization.

### 3.2 Data Augmentation

We propose a way to enhances the model's robustness to explicit rules by increasing its exposure to rule-following data points, thereby enabling the model to generalize the rules across the entire input space. We achieve this through a data augmentation component that generates augmented data samples conforming to the rules and trains the model on a combination of original and augmented data. Specifically, for any input $x$ expected to satisfy a rule $r = (c, \mathcal{P})$, we generate an augmented input $x'$ such that $\texttt{MATCH}_x(x', c)$ holds true. One such strategy for finding $x'$ is detailed in Algorithm 1, which is employed throughout our experiments. Alternatively, neighborhood perturbation techniques can be applied, where $x$ is perturbed by $\delta$, ensuring that $x' = x + \delta \implies \texttt{MATCH}_x(x + \delta, c)$.

Data augmentation occurs during the batch pre-processing stage (see Figure 1(c)) and dynamically adjusts the training data mixture using a parameter $\alpha$, formulated as:

$$X_\alpha = \alpha X' + (1 - \alpha)X$$

A carefully balanced mixture allows the model to address both the application-specific task and ensures compliance with the defined rules.

By doing this transformation, we can generate augmented data just-in-time, and also incorporate expert domain knowledge in doing so. Further, how much counterfactual data to augment, and what loss gets

backpropagated is controlled by the regularization module. We refer to counterfactual data augmentation inspired by prior work in causal inference (Pearl & Mackenzie, 2018), but under a restrictive independence assumption that changes to a single input feature using the *do* operator, does not interfere with other input features, and measure the faithfulness of the model's outputs to the domain-specific rules that are narrowly defined as mapping between individual input and output concept classes. Modeling interactions between a set of inputs and output is currently out of scope in our problem formulation.

### 3.3 Regularization

For a given rule $r = (c, \mathcal{P})$, data augmentation constrains the augmented input $x'$ to the concept class $c$ and samples the output category $c_y$ according to the distribution $\mathcal{P}$ (see Algorithm 1). However, to ensure that the model $\mathcal{M}$ follows the rule $r$ (in expectation), we decompose it into deterministic hard rules, mapping $c$ to $c_y$, and transform these discrete statements into a regularization loss defined over a continuous representational space.

For instance, let $|C_y| = k$, where $k$ represents the number of categories in $C_y$. We learn a mapping from the model's penultimate layer $\mathcal{M}'(x)$ using a feed-forward layer $h$, which is connected to a softmax function with $k$ units. (The original model for the primary task can be formulated as $\mathcal{M}(x) = g(\mathcal{M}'(x))$, with $g$ being a task-specific feed-forward layer.) We train this feed-forward layer $h$ on the original training data and their corresponding concept classes $c_y$ using a cross-entropy loss:

$$\forall x \in X : \mathcal{L}_{ce}(h(\mathcal{M}'(x)), c_y)$$

This learned layer then encodes the output membership function and is subsequently used to enforce the functional constraint that the output belongs to a specific concept class over the augmented data:

$$\mathcal{L}_{reg}(x', c_y) = \mathcal{L}_{ce}(h(\mathcal{M}'(x')), c_y)$$

For a given rule, $r = (c_x, \mathcal{P})$, we also experiment with the way in which the corresponding penalty terms are incorporated into the base loss function $\mathcal{L}_{app} = \mathcal{L}_{ce}$, using a parameter $\beta$, defined as:

$$\begin{aligned}(1 - \alpha)\mathcal{L}_{app}(g(M'(x)), y) + \alpha[\beta\mathcal{L}_{reg}(x, c_y) \\ + (1 - \beta)\mathbb{E}_{x' \sim C2I(c_x); c'_y \sim \mathcal{P}}\mathcal{L}_{reg}(x', c'_y)]\end{aligned} \tag{1}$$

where $c_y \sim O2C(y)$, and $x'$ is sampled from a uniform distribution of concepts $C2I(c_x)$ (See Figure 1(a) and Figure 1(b)). These parameters $(\alpha, \beta)$ are optimized via Bayesian optimization to maximize rule adherence and model performance over a held-out validation set. We call this learning paradigm as the Domain Faithful Deep Learning ( `DFDL`) framework.

---

**Algorithm 1** `DFDL` Framework

---

1: Sample $(x, y) \sim D$
2: Sample $c_x \sim \text{Uniform}(I2C(x))$
3: Sample $c_y \sim \text{Uniform}(O2C(y))$
4: Sample $r = (c, \mathcal{P}) \sim \text{Uniform}(C2R(c_x))$
5: Sample $x' \sim \text{Uniform}(C2I(c_x))$
6: Sample $c'_y \sim \mathcal{P}$
7: Propagate the loss Eqn (1)

---

> **Definition of domain faithfulness** We say that a classifier $f$ is domain faithful with respect to $\mathcal{P}$ over a data distribution $(x, y)$ if for all perturbations $x' : c_x = c_{x'}$, the domain constraints $\mathcal{P}$ are satisfied.

In conclusion, data augmentation involves enriching the dataset with examples that are explicitly designed to satisfy or test the domain rules, ensuring that the model is exposed to instances where the rules are relevant. Regularization, on the other hand, introduces a penalty in the loss function to encourage the model $\mathcal{M}$ to satisfy the domain rules, even if the rules are not perfectly differentiable. This decomposition allows for the indirect enforcement of domain rules while maintaining compatibility with standard differentiable optimization procedures used in deep learning.

### 3.4 Parameter Optimization

We optimize all parameters in $g, h, \mathcal{M}'$, in addition to hyper-parameters: $\alpha, \beta$. This allows the domain experts to train domain specific end-to-end neural networks by simply specifying the parameters of their constraints.

We optimize the hyperparameters using techniques of Bayesian optimization (Martinez-Cantin, 2014) which models the prior of the values based on the domain expert assigned values and evaluates based on an additive semiparametric error $\epsilon(x)$ to compute the posterior of the surrogate model $P(\alpha, \beta | D)$ given an initial distribution of functions (a Gaussian process) and a criterion to acquire new values. In our empirical Bayes implementation, we use the acquisition function based on a combination of global and local derivative-free method - BOBYQA (Powell, 2009).

## 4 Theoretical insights

### 4.1 Background and Notations

We consider the case where we aim to learn hypothesis $h : \mathcal{X} \to \mathcal{Y}$ in the hypothesis class $\mathcal{H}$, which learns to predict the label $y$ for input $x$ of dimension $d$. We aim to minimize the errors in prediction, i.e. the indicator function $\mathbb{1}(h(x) \neq y)$ (w.l.o.g, we can substitute any loss function such as cross-entropy, here).

**Empirical risk minimization**: Given a distribution $\mathcal{D}$, and $N$ examples $(x_i, y_i)$ drawn i.i.d from $\mathcal{D}$, the hypothesis that minimizes the empirical risk minimization objective is given by

$$min_{h \in \mathcal{H}} \frac{1}{N} \sum_{i=1}^{N} \mathbb{1}(h(x_i) \neq y_i) \tag{2}$$

$$\tag{3}$$

**Adversarial Robustness**: The adversarial robustness counterpart Goodfellow et al. (2014) for the above empirical risk minimization is to assume a set of allowed perturbations of the input $\tau \subset R^d$, for which the perturbed input $x + \delta$ where $\delta \in \tau$, is still mapped to the same label $y$. In this case, the hypothesis that minimizes the adversarial risk minimization objective is given by

$$min_{h \in \mathcal{H}} \mathbb{E}_{(x,y)}[sup_{\delta \in \tau} \mathbb{1}(h(x + \delta) \neq y)] \tag{4}$$

$$\tag{5}$$

**Probabilistic robustness**: The probabilistic counterpart (Robey et al., 2022) of the above adversarial robustness objective is to minimize the probability that the perturbed input $x + \delta$ is not mapped to the label $y$. Hence, the hypothesis that minimizes the probabilistic robustness objective for a given error probability level $\rho$ is given by

$$min_{h \in \mathcal{H}} \mathbb{E}_{(x,y)}[\mathbb{1} \; \mathbb{P}_{\delta \in \tau}[(h(x + \delta) \neq y)] > \rho] \tag{6}$$

$$\tag{7}$$

### 4.2 Statistical Properties

We first present that the hypothesis class for domain faithful models is a strict subset of the probabilistically robust models as proposed in (Robey et al., 2022).

**Proposition 1**: Specifically, for any data point $(x, y)$, for every domain faithful model such that $f(I2C(x + \delta)) = O2C(y)$, there is a canonical $(\tau, \rho)$ probabilistically robust model $h^*$,: $h^* = min_{h \in H} \mathbb{E}_{(x,y)}[\mathbb{I}[P_{\delta \sim \tau}((h(x + \delta) \neq y) > \rho]]$. Here, $\tau$ is the set of allowed perturbations on x, which will keep the input class the same.

Proof sketch: We find that in our case, $\tau$ defined by the domain specific constraints in the input, i.e. $I2C(x + \delta) = I2C(x)$, is a strict subset of $||\delta||_2 < \epsilon$, where $\epsilon$ is the perturbation in a manifold where $x$ is embedded such that all inputs belonging to the same category $I2C(x)$ is embedded within a maximum

Euclidean distance of $\epsilon$. One could achieve such an embedding by transforming the set of input category rules $I2C(x)$ to cluster $x$ where the maximum intra-cluster distance is bounded by $\epsilon$: $max_\delta[I2C(x) = I2C(x+\delta)] < \epsilon$. Further, we see that the condition $h(x + \delta) \neq y$ can be relaxed as $f(I2C(x + \delta))) \neq O2C(y)$.

**Corollary 1a**: The sample complexity for domain faithful models $\Theta(H) < log(\frac{1}{\rho_0})\Theta(1)$ where $\Theta(1)$ is sample complexity to optimize empirical risk minimization hypotheses [Prop 4.2 of (Robey et al., 2022)]

**Corollary 1b**: The risk gap between Bayes optimal $h^*_{Bayes}$ and domain faithful model $h^*_{(\tau,\rho)}$ is $\mathcal{O}(\frac{1}{\sqrt{d}})$, where $d$ is the number of dimensions of $x$. [Prop 4.1 of (Robey et al., 2022)] (proof in Appendix)

Further, we prove the convergence properties of our proposed domain faithful model by demonstrating equivalence between Eq. (1) and a combination of multitask and adversarially robust learning. Let $|||.|||^2_r$ denote the squared Schatten norm, $\hat{r} = \frac{2r}{r+1}$ for $r \geq 1$, and $tr(\Omega)$ denote the trace of the task covariance matrix between $g, h$ in Eq. (1).

**Theorem 4.1.** *If $tr(\Omega) < q_1$ and $|||\theta|||^2_{\hat{r}} < q_2$ for some fixed constants $q_1, q_2$, then minimizing Eq. (1) through projected gradient descent learns domain faithful model parameterized by $\theta$ where after $k+1$ steps, the violation rate is non-increasing $\epsilon_{k+1}(\theta) \leq \epsilon_k(\theta)$.*

**Proof sketch**: To satisfy domain faithfulness in the concept space, where for $(x, y)$, $x+\delta : I2C(x) = I2C(x+\delta)$ would preserve the property that $f(I2C(x + \delta)) = O2C(y)$. Using prior results in adversarial robustness, we can then formalize this as finding the parameter $\theta$ as follows, where S denotes the perturbations in a t-norm space $\delta : ||\delta||_t < \epsilon$:

$$\min_\theta \mathbb{E}_{(x,y)\sim D}[max_{\delta\in S}\mathcal{L}_{reg}(x + \delta, O2C(y))] \tag{8}$$

For an adversarially trained model upon $k$ projected-gradient descent steps, adversarial perturbations can be obtained by

$$\delta^{k+1} = proj_S(\delta^k + \eta\, sign(\nabla_{\delta^k}\mathcal{L}_{reg}(x + \delta, O2C(y))))$$

Since our perturbations are in the discrete concept space, our adversarial training can be stopped when $\forall x, I2C(x + \delta^{k+1}) = I2C(x)$. Since computing the maximum over the space $S$ is intractable, we can further relax this using a sum over the samples of perturbations and perform batched gradient descent. The proof then follows from Theorem 1 from (Zhang et al., 2018a) and Corollary A.2 of (Madry, 2017). We provide the full proof below.

## 4.3 Proofs

We first present that the hypothesis class for domain faithful models is a strict subset of the probabilistically robust models as proposed in (Robey et al., 2022).

**Proposition 1**: Specifically, for every domain faithful model such that $f(I2C(x + \delta)) = O2C(y)$, there is a canonical $(\tau, \rho)$ probabilistically robust model $h^*$: $h^* = min_{h\in H}\mathbb{E}_{(x,y)}[\mathbb{I}[P_{\delta\sim\tau}((h(x + \delta) \neq y) > \rho]]$

We provide the proof by construction. For the domain faithful model such that $f(I2C(x + \delta)) = O2C(y)$, we define $h = f \cdot I2C$. Further, we transform the input domain $(x + \delta) \rightarrow (x' + \delta')$ such that $I2C(x + \delta) = I2C(x) \rightarrow ||\delta'||_2 < \epsilon$. We obtain such an input transformation by construction where $x' \in \mathbb{R}^d$, where $d = d_X + |C_x|$ where $d_X$ is the dimension of $x \in X$, and $|C_x|$ denotes the number of input categories - ordered as $\{c_1, c_2, ..\}$. We achieve the necessary property that the Euclidean distance between inputs in the same input category is bounded by $\epsilon$: $dist(x'_1, x'_2) < \epsilon$ if $I2C(x_1) = I2C(x_2)$ by constructing $x'$ as follows: $x'[i] = \frac{\epsilon}{\sqrt{d_X}x_{max}[i]}x[i]$ if $i \leq d_X$ and $x'[i] = K$ if $I2C(x) = c_{i-d_X}$ and $x'[i] = 0$ otherwise.

Thus,

$$dist(x_1', x_2') = \sqrt{\frac{\epsilon^2}{d_X} \sum_{i=1}^{d_X} \frac{(x_2'[i] - x_1'[i])^2}{x_{max}[i]^2} + \mathbb{1}[I2C(x_1) \neq I2C(x_2)]2K^2}$$
$$< \sqrt{\epsilon^2 + \mathbb{1}[I2C(x_1) \neq I2C(x_2)]2K^2}$$
$$< \epsilon \text{ if } \mathbb{1}[I2C(x_1) = I2C(x_2)]$$

Thus, $\delta' \sim \tau$ defined by the domain specific constraints in the transformed input, i.e. $I2C(x + \delta) = I2C(x)$, is a strict subset of $||\delta||_2 < \epsilon$, where $\epsilon$ is the perturbation in a manifold where $x$ is embedded such that all inputs belonging to the same category $I2C(x)$ is embedded within a maximum Euclidean distance of $\epsilon$. Further, we see that $f(I2C(x + \delta))) \neq O2C(y)$ can be mapped to $P[h(x' + \delta') \neq y'] > \rho$ by constructing a random variable $y' = O2C(y)$ with probability $1 - \rho$.

The following corollaries hold directly from the properties of probabilistically robust learning Robey et al. (2022).

**Corollary 1a**: The sample complexity for domain faithful models $\Theta(H) < log(\frac{1}{\rho_0})\Theta(1)$ where $\Theta(1)$ is sample complexity to optimize empirical risk minimization hypotheses [Prop 4.2 of (Robey et al., 2022)]

**Corollary 1b**: The risk gap between Bayes optimal $h_{Bayes}^*$ and domain faithful model $h_{(\tau,\rho)}^*$ is $\mathcal{O}(\frac{1}{\sqrt{d}})$, where $d$ is the number of dimensions of $x$. [Prop 4.1 of (Robey et al., 2022)] (proof in Appendix)

Further, we prove the convergence properties of our proposed domain faithful model by demonstrating equivalence between Eq.(1) and a combination of multitask and adversarially robust learning. Let $|||.|||_r^2$ denote the squared Schatten norm, $\hat{r} = \frac{2r}{r+1}$ for $r \geq 1$, and $tr(\Omega)$ denote the trace of the task covariance matrix between $g, h$ in Eq.(1).

**Theorem 4.2.** *If $tr(\Omega) < q_1$ and $|||\theta|||_{\hat{r}}^2 < q_2$ for some fixed constants $q_1, q_2$, then minimizing Eq. (1) through projected gradient descent learns domain faithful model parameterized by $\theta$ where after $k+1$ steps, the violation rate is non-increasing $\epsilon_{k+1}(\theta) \leq \epsilon_k(\theta)$.*

Learning $f$ and $\hat{p}$ involves a multi-task learning between $(f(x), y)$ and $(\hat{p}(x), C_Y(y))$. This task can be generalized into the following task as proposed in (Zhang et al., 2018a), where $f$ is parameterized by a weight matrix $\theta$ trained over $n$ data points, a regularizer $g$, and hyperparameters $\lambda_1, \lambda_2$:

$$\min_{\theta, \Omega} \begin{bmatrix} \frac{1}{2n} \sum_{i=1}^n \mathcal{L}_{ce}(f_\theta(x_i), y_i) \\ + \mathcal{L}_{ce}(C_Y(f_\theta(x_i)), C_Y(y_i)) \\ + \frac{\lambda_1}{2} tr(\theta\Omega^{-1}\theta^T) + \frac{\lambda_2}{2} g(\Omega) \end{bmatrix} . \tag{9}$$

Under the assumption that $g(\Omega) = tr(\Omega^r)$ for any positive scalar $r$, and by our assumptions, and noting that $\theta$ parameterizes both the tasks in our case, we can further reduce the problem to

$$\min_\theta \begin{bmatrix} \frac{1}{2n} \sum_{i=1}^n \mathcal{L}_{ce}(f_\theta(x_i), y_i) \\ + \mathcal{L}_{ce}(C_Y(f_\theta(x_i)), C_Y(y_i)) + \lambda_1 |||\theta|||_{\hat{r}}^2 \end{bmatrix} \tag{10}$$

To satisfy domain faithfulness in the concept space, where for $(x, y)$, $x + \delta : I2C(x) = I2C(x + \delta)$ would preserve the property that $f(I2C(x + \delta)) = O2C(y)$. Using prior results in adversarial robustness, we can then formalize this as finding the parameter $\theta$ as follows, where S denotes the perturbations in a t-norm space $\delta : ||\delta||_t < \epsilon$:

$$\min_\theta \mathbb{E}_{(x,y) \sim D}[max_{\delta \in S} \mathcal{L}_{reg}(x + \delta, O2C(y))] \tag{11}$$

For an adversarially trained model upon $k$ projected-gradient descent steps, adversarial perturbations can be obtained by

$$\delta^{k+1} = proj_S(\delta^k + \eta \, sign(\nabla_{\delta^k} \mathcal{L}_{reg}(x + \delta, O2C(y))))$$

Since our perturbations are in the discrete concept space, our adversarial training can be stopped when $\forall x, I2C(x + \delta^{k+1}) = I2C(x)$. Since computing the maximum over the space $S$ is intractable, we can further relax this using a sum over the samples of perturbations and perform batched gradient descent. The proof then follows from Theorem 1 from (Zhang et al., 2018a) and Corollary A.2 of (Madry, 2017).

## 5 Implementation Specifications

In this section, we provide a more detailed analysis of the applications referenced in the main body of the paper. Due to constraints of space and the diverse nature of the applications, we were unable to include an in-depth discussion within the paper itself. While we have included a summary table to address this limitation, fully understanding the intricacies of each application may require additional elaboration.

Table 1: Detailed description of domain rules for each application

|  | Synthetic Predicates | Medical Recommendation | Box2D Car Racing | Toxicity Detection |
|---|---|---|---|---|
| $\mathcal{X}$ | $[-1, 1]$ | Set of Diagnostic (ICD-9) codes | Set of all possible states of the racing car | Comments |
| $\mathcal{Y}$ | $[-1, 1]$ | Set of medication (ACT) codes | Actions that the car can take, i.e., 7 steering options mentioned in section | {Hateful, Non-Hateful} |
| $C_x$ | $[-1, 1]$ | Set of Diagnostic (ICD-9) codes | Set of the state of the car that runs off the track | Comments with cuss words |
| $C_y$ | $[-1, 1]$ divided into $k$ buckets | Set of medication (ACT) codes | Actions that the car can take, i.e., 7 steering options mentioned in section | {Hateful, Non-Hateful} |
| **Rules** | Mapping the input concept class to the bucket in which the output lies, as per the predicate selected. | Co-occurrences obtained from Weisser et al. (2008) between each ICD-9 code and ACT code | Mapping states that lead to non-optimal outcomes | Comments with cuss words are hateful comments |
| **Data Aug** | By sampling more data points outside the predicate domain | By substituting a diagnosis with another diagnosis of the same category | By using a PID controller to generate non-optimal trajectories | By adding cuss words in non-hateful comments |
| $\mathcal{L}_{reg}$ | Misclassification | Misclassification | Penalty imposition on non-optimal states | Misclassification |
| **Comments** | Allows generalization of rules outside of the provided domain | Ensures that diagnoses of the same category are provided with medicine of a similar category | This augmentation leads to negative trajectories, and the model is penalized for exploring them | Leads to improvement of performance in toxicity classification on an adversarial benchmark |

For each application, we examine five key aspects:

1. **The Task:** What is the specific problem being addressed, and what are the objectives of the task?

2. **The Rules:** What are the domain-specific rules or guidelines that must be adhered to?

3. **Implementation Details:** How were the solutions for each task executed and implemented?

4. **Performance Sensitivity Discussion:** How does enforcing adherence to the rules impact the overall performance of the task?

5. **Notion of Domain Faithfulness:** To what extent were we able to enforce the domain-specific rules in a manner faithful to the original task?

We hope that this structure enables a comprehensive understanding of each application and its associated challenges.

## 5.1 Synthetic Predicates

### 5.1.1 The Task

We adopt the task from (Tavares et al., 2019), where pairs $(x, y)$ are jointly sampled uniformly from the interval $[-1, 1]$, subject to a predicate $p(x, y)$, which the values of $x$ and $y$ must satisfy. This process generates the synthetic dataset of $(x, y)$ pairs. The goal is to train a deep neural network (DNN) on this dataset to predict $y$ given an input $x$.

### 5.1.2 The Rules

The *predicates act as the defining rules* for the dataset, but they are only provided for a smaller range of values $X_{dom(C)} \in X$, specifically within the interval $[-0.1, 0.1]$. Consequently, the model is expected to faithfully predict $y$ by learning the underlying rules that hold for the full range $[-1, 1]$. The evaluation is based on the following three predicates:

- $x = y$
- $|x| \geq |y|$
- $x^2 = y^2$
- $\sin(kx) \cdot \cos(kx) \geq 0.9999$

### 5.1.3 Implementation Details

The rules are integrated within the framework provided by (Tavares et al., 2019). The input space $X$ is represented by a single concept class $C_x$, with every element in the interval $[-1, 1]$ adhering to this concept. The output space $y$ is discretized into $k$ consecutive continuous bins, and these bins constitute the output categories in $C_y$. The rules define mappings from the input concept class $C_x$ to the categories in $C_y$, as determined by the predicates. In this case, the predicates are defined as the mean of the values of the predicates to be learned for a given grid value of the input $c_x = I2C(x)$, such that $f(c_x) = \mathbb{E}_{x' \sim C2I(c_x)} p(x')$.

The details of the baseline model are provided in the main body of the paper. To evaluate the performance of the model, we employ the Root Mean Squared Error (RMSE) metric, which measures the discrepancy between the model's predictions and the ground truth $y$ values. We train all the feedforward models for 100 epochs on Keras platform with 1 NVIDIA RTX8000 GPU.

### 5.1.4 Performance Sensitivity

We find that the performance gains is most observed in non-linear predicates ($sin(kx)cos(kx) \geq 0.999$) than linear and polynomial predicates ($x = y, x^2 = y^2$. Further, the reduction in RMSE scales with the size of the grid used to define the domain-rules. We note that asymptotically, a grid size of infinitesimal size corresponds to the predicate itself.

### 5.1.5 Domain Faithfulness

Further, we find that any noise in the domain rule specification increases the RMSE, with only 10% rule noise enough to produce RMSE worse than the baselines. This further demonstrates the need to incorporate accurate and well-founded domain rules in our framework.

## 5.2 Medical Recommendation

### 5.2.1 The Task

This research addresses a recommendation task using the MIMIC-III dataset (Shang et al., 2019), focusing on recommending a set of medications (represented by ATC codes) for a patient's diagnosis (represented by ICD-9 codes). The goal is to develop a neural network model that, given a sequence of ICD-9 codes as input (reflecting a patient's diagnostic history), generates a candidate set of ATC codes, i.e., medications, as output.

### 5.2.2 The Rules

To enhance the model's predictions, we incorporate expert priors on the co-occurrences between diagnostic codes ($x$, ICD-9) and medication codes ($y$, ATC), as detailed in (Weisser et al., 2008). The dataset contains a validated statistical table based on pairwise mutual information scores reflecting these co-occurrences. This expert knowledge serves as a guiding principle in the form of rules that the model must adhere to. Specifically, *for similar diagnoses, it is expected that the corresponding medications should be similar*. We aim to encode this expectation by utilizing the expert co-occurrence matrix to influence the model's predictions.

### 5.2.3 Implementation Details

Formally, for each diagnosis $d \in C_x$ (where $C_x$ denotes the set of ICD-9 diagnosis codes) and medication $m \in C_y$ (where $C_y$ represents the set of ATC medication codes), we have the co-occurrence score $\texttt{score}(d, m)$ using the expert co-occurrence matrix. For each diagnosis $d$, a rule $r = (d, \mathcal{P})$ is defined, where $\mathcal{P}$ represents a distribution over medications, derived by normalizing the scores in the matrix. Specifically, unnormalized $\mathcal{P}(m = m')$ is given by $\texttt{score}(d, m')$ for all $m' \in C_y$.

In our experiment, we collapse $\mathcal{P}$ into a point mass, transforming the rule into an *exact rule $r = (d, m^*)$*, where $m^*$ is the medication that maximizes the co-occurrence score, i.e., $m^* = \arg\max_{m'} \texttt{score}(d, m')$ for all $m' \in C_y$.

Once the $\texttt{RULES-SPACE}$ is defined, the model is trained using the $\texttt{DFDL}$ algorithm as described in the main paper. The baseline G-BERT model is pretrained on single visit MIMIC data, with an input embedding size of 75 with 4 attention heads in the transformer architecture. We used 2 hidden layers, with 300 hidden dimensions and 4 attention heads per layer. We train on 100 epochs, with learning rate of $5e^-4$, L1 norm of 1.1, and dropout probability of 0.4.

### 5.2.4 Performance Sensitivity

We find that the relative performance gains are highest for patients with diagnostic codes that are least frequent in the training dataset and mapped by the domain specific rules, while lowest for diagnostic codes not covered by the domain rules.

### 5.2.5 Domain Faithfulness

Rules that have least stochasticity are also the ones that have the least violation rates, whereas the rules for diagnostic categories which can map to multiple medication categories are easily violated even in our best-performing models. This is an area of improvement for future work that needs to be addressed beyond data augmentation and regularization. The issue stems from the inability to distinguish between noisy rules and rules with inherent stochasticity as we model each patient history independently.

## 5.3 Box2D Car Racing

### 5.3.1 The Task

The objective of this work is to develop a deep neural network (DNN) model capable of controlling a 2D car in OpenAI's Gym task, Box2D CarRacing (Brockman et al., 2016). The task requires the car to navigate a track efficiently, reach the finish line, and avoid deviating from the path. The input to the neural network

consists of the car's 2D top-down view in the environment, along with its velocity and sensor readings. These inputs are preprocessed by flattening them into a one-dimensional array, which is then fed into the neural network. The network outputs the actions the agent should take, where each action corresponds to the one that maximizes the Q-value, guiding the car's behavior.

### 5.3.2  The Rules

We want to ensure the following general rule that the *car remains on track*. To enforce this rule, we sought to go beyond the environment's default penalty system, which imposes a negative reward of -100 when the car goes off track. While this built-in penalty helps steer the agent's behavior, it introduces sparse feedback, as the penalty is only triggered after the violation occurs. To address this, we aimed to incorporate this constraint directly using data augmentation which generates more such scenarios and helps the integrate within the deep learning framework through empirical risk minimization. By doing so, we ensure the car stays on the track more consistently without relying solely on the delayed off-track penalty, allowing for more immediate and continuous guidance during training.

### 5.3.3  Implementation Details

In this context, each rule is applied to the state of the car, though the approach used here differs from the ones described previously. In prior applications, we sought to enforce rules as desired behavior. Here, we are introducing rules to prevent specific behaviors—namely, the car going off the track. This can be interpreted as enforcing the opposite of undesired behavior, maintaining consistency in rule application, even though it may initially seem different. To achieve this, we augment the training data by introducing states that lead the car off the track, akin to training on negative examples to improve robustness.

In this setup, $C_x$ represents a single concept $S$: the states from trajectories that lead to the car leaving the track. Meanwhile, $C_y$ corresponds to the original action space $\mathcal{Y}$, which defines the possible actions in the environment.

We employ *generative data augmentation*, where we explicitly construct trajectories that lead the car off the track using a Proportional-Integral-Derivative (PID) controller. This allows the model to learn from these off-track states, enhancing its ability to avoid similar situations in the future.

The rule $r = (S, \mathcal{P})$, where $\mathcal{P}$ is the distribution over actions that lead to the car going off the track. By introducing these states and action distributions into the training process, the model learns to recognize and avoid risky trajectories, leading to more robust behavior during deployment.

### 5.3.4  Performance Sensitivity

We find that gains in early rewards mostly stems from the RL model learning to steer and accelerate alternatively in turns which provides avoids the car from going off-track, the domain rule that was enforced. While this can lead to unstable with alternative left and right maneuvering, and should be addressed in future work with additional domain rules.

### 5.3.5  Domain Faithfulness

A high negative reward for the car going off-track we saw could be prohibitive for learning and sometimes led to random trajectories. To mitigate this, we introduced the domain specific rule only after the first 10 frames of the game. This helped the dynamics of the car controls to stabilize before learning to hillclimb on the car racing rewards.

## 5.4  Toxicity Detection

### 5.4.1  The Task

The task of toxicity detection involves developing a text classification model that can identify hateful comments on online platforms. The model is trained to classify sentences as either hate or non-hate. Given an input

sentence $s \in \mathcal{X}$, the model outputs a set of scores corresponding to various toxicity categories, including "Toxic," "Obscene," "Threat," "Insult," and "Hateful." Our focus is specifically on the identification of hateful sentences.

Traditional models in this domain are often constrained by the limitations of their training datasets, leading to misclassification of new or nuanced forms of toxic content. This is particularly problematic in high-stakes scenarios where errors in detecting hate speech can have significant consequences. In such cases, practitioners may adopt a more conservative approach, where the model is designed to prioritize minimizing false negatives (i.e., failing to detect hateful content) over false positives (i.e., incorrectly labeling non-hateful content as hate).

### 5.4.2   The Rules

To enhance the classifier's ability to identify toxic content, domain-specific rules are introduced through a curated list of curse words. The goal is to ensure that the *model flags any sentence as hateful if it contains a curse word,* particularly in cases where the sentence was originally classified with high confidence as non-hateful by the model.

### 5.4.3   Implementation Details

In this context, the set $C_x$ consists of a single concept, $H$, which represents the collection of sentences containing hateful curse words. To generate additional sentences that adhere to the $C_x$ category, we construct new sentences by inserting a curse word into sentences that the model classifies as non-hateful with high confidence. The output space, denoted by $C_y$, is the same as $\mathcal{Y}$, consisting of the classes "hateful" and "non-hateful."

The rule is defined as $r = (H, \mathcal{P})$, where $\mathcal{P}(y = \text{hate})$ is a probability distribution over the output classes, conditioned on the presence of any curse word from the curated list in the sentence $s$. To simplify, we collapse $\mathcal{P}$ into a single class, "hateful," resulting in an *exact rule* that mandates the classification of any sentence containing a curse word as hateful.

Again, once the `RULES-SPACE` is defined, the model is trained using the `DFDL` algorithm as described in the main paper.

The model used for this task is PerspectiveBERT, a state-of-the-art (SOTA) text classifier. This is pretrained on online comments from public sources like Wikipedia and news sources like the New York Times. The pre-trained model was then fine-tuned to predict the toxicity levels as determined by the fraction of human raters who found a sentence to be toxic. The number of human raters per task varied from 3-10.

### 5.4.4   Performance Sensitivity

We find the performance gains mostly are from the reduction in false negative rates in predicting the 5 class labels (toxic, obscene, hate, threat, and insult), with minimal increase in the false positive rates. This further demonstrates that since the nature of the data being augmented can significantly improve the performance of the model in a fine-grained manner. However, future work should focus on improving performance on adversarial datasets like Dynabench Kiela et al. (2021).

### 5.4.5   Domain Faithfulness

As previously noted, there was no gains in performance due to the reduction of false positive rates, which would require additional rules on data augmentation trying to capture false signals of toxicity, like slang, regional dialects, etc. Incorporating such rules with the help of expert linguistics can be pursued in future work. Further, future work can address fairness issues in toxicity classifiers by incorporating external knowledge and secondary attributes can further enhance the performance of the model Balashankar et al. (2021b)

Table 2: Our method considerably reduces the RMSE to the ground truth on average over the 4 synthetic conditioned models.

| Model Version | RMSE to ground truth |
|---|---|
| Baseline | 0.84 (0.80, 0.88) |
| Baseline+Mapped | 0.75 (0.74, 0.76) |
| Baseline+Predicate | 0.72 (0.68, 0.76) |
| DeepStochLog Winters et al. (2022) | 0.65 (0.60, 0.71) |
| Logic Distillation Hu et al. (2016) | 0.68 (0.63, 0.72) |
| RA | 0.65 (0.63, 0.68) |
| RA-WCR | 0.42 (0.41, 0.43) |
| RA-WCR [dfdl] | **0.28 (0.21, 0.35)** |

## 6 Evaluation

We evaluate the models using our framework on 1 synthetic dataset, and 3 real datasets. The domain-specific rules are drawn from prior work on deriving inductive biases in these domains. For the following domains, we define the current state-of-the-art model as ***Baseline***. As our framework incorporates more information through robust domain specific mappings through counterfactual augmented data, we also developed additional baselines that used these priors as input features. Specifically, we augmented categorical embeddings of each input to form the ***Baseline*+Cat** model. In this baseline, no expert validation information is provided, but the category embedding is explicitly provided. We also augmented the embeddings of the applicable rule-based output category $c_y$ (for ***exact*** rule $r = (c_x, c_y)$) as an input to the model to form the ***Baseline*+Mapped** model. This trains the model to pay attention to the mapped output category and minimize category misclassification. We also incorporate the predicate exchange algorithm (Tavares et al., 2019) to incorporate the categorical rules as stochastic approximations which is then fine-tuned using a temperature-scaled MCMC to draw posterior samples for a probabilistic program execution. Finally, we instantiate our models ***Baseline* RA**, which modifies the baseline with Rule-based Augmentation ($\beta = 1$ in Eq. 1) and ***Baseline* RA-WCR**, which uses Rule-based Augmentation and Within-Category Regularization ($\beta < 1$). Our domain faithful deep learning version implemented in JAX (jaxverify) is called ***Baseline* RA [dfdl]** and ***Baseline* RA-WCR [dfdl]**. We set $\alpha = 0.2$ after cross-validation (more details in Appendix). We also compare with other methods from the literature that incorporate rules such as Hu et al. (2016) and Winters et al. (2022), with the same ***Baseline*** as the backbone for fine-tuning.

### 6.1 Synthetic Predicates

We use the histograms of samples from a uniform prior $[-1, 1]^2$ used in (Tavares et al., 2019), conditioned on a variety of predicates. These examples are simple yet challenging to simple feed-forward neural networks to learn due to discontinuities in the approximate posterior. Specifically, the true distribution is from $x, y \sim Unif(-1, 1)$ with conditions of $x = y$, $|x| \geq |y|$, $x^2 = y^2$ and $sin(kx).cos(kx) \geq 0.9999$. In each of these cases, we are given a partial domain knowledge in the space of $x, y \sim Unif(-0.1, 0.1)$ where each of the conditions and the corresponding constraints are known to be true. Through our approach of rule-based augmentation and regularization, we show that we are able to learn more accurate models as shown in Table 2. Baseline is a feed-forward network model with two layers with 10 hidden units with non-linear units (RELU) run for 100 epochs.

### 6.2 Medication Recommendation

We follow the MIMIC-III medication recommendation task as per (Shang et al., 2019), and the domain specific mappings $p$ are obtained from (Weisser et al., 2008) where medical experts validated a statistical table based on pairwise mutual information scores of co-occurrences between diagnostic $x$ (ICD-9) and medication $y$ (ATC) codes. These validated tables are segmented based on the age and gender of Austrian patients. Note that this dataset is different from the MIMIC-III dataset used in our evaluation. Hence, we use only the pairs of ICD-9, ATC categories that are expert validated, but not any other statistical information from this

Table 3: Our RA-WCR model improves accuracy metrics of G-BERT on the MIMIC-III medication recommendation task after fine-tuning the parameters of the constraints for the Original dataset and the category classification task for the within-category Augmented dataset.

| | Model | Jaccard | F1 | PR-AUC |
|---|---|---|---|---|
| Original | G-Bert | 0.3679 ±0.01 | 0.5281 ±0.03 | 0.6212 ±0.03 |
| | G-Bert+Cat | 0.3564 ±0.02 | 0.5203 ±0.04 | 0.6146 ±0.03 |
| | G-Bert+Mapped | 0.3680 ±0.01 | 0.5299 ±0.03 | 0.6230 ±0.02 |
| | G-Bert+Predicate | 0.3470 ±0.01 | 0.5149 ±0.03 | 0.6291 ±0.02 |
| | G-Bert RA | 0.3883 ±0.02 | 0.5788 ±0.02 | 0.6541 ±0.01 |
| | G-Bert RA-WCR | 0.4300 ±0.01 | 0.5967 ±0.01 | 0.6775 ±0.02 |
| | G-Bert RA-WCR [dfdl] | **0.4530** ±0.01 | **0.6132** ±0.01 | **0.6802** ±0.02 |
| Augmented | G-Bert | 0.3677 ±0.03 | 0.5281 ±0.02 | 0.6199 ±0.00 |
| | G-Bert+Cat | 0.3301 ±0.03 | 0.5102 ±0.01 | 0.5952 ±0.01 |
| | G-Bert+Mapped | 0.3573 ±0.01 | 0.5249 ±0.02 | 0.6084 ±0.02 |
| | G-Bert+Predicate | 0.3564 ±0.01 | 0.5223 ±0.02 | 0.6051 ±0.02 |
| | G-Bert RA | 0.3723 ±0.02 | 0.5483 ±0.02 | 0.6343 ±0.01 |
| | G-Bert RA-WCR | 0.4033 ±0.01 | 0.5699 ±0.02 | 0.6596 ±0.02 |
| | DeepStochLog | 0.3592 ±0.03 | 0.5195 ±0.02 | 0.6138 ±0.02 |
| | Logic Distillation | 0.3865 ±0.02 | 0.5692 ±0.01 | 0.6417 ±0.01 |
| | G-Bert RA-WCR [dfdl] | **0.4127** ±0.01 | **0.5742** ±0.02 | **0.6621** ±0.02 |

study. A total of unique 349 pairs of ATC and ICD-9 Level 2 codes were deemed to be valid by the experts; 958 unique pairs if we break down by age and gender forms our domain specific mapping $f$. Age is bracketed into 3 ranges based on year of birth (1949-68, 1969-88, 1989-2008) and gender is considered to be binary (male, female).

We use the current state-of-the-art for the medication recommendation task on MIMIC-III dataset as the *Baseline* - G-BERT (Shang et al., 2019). This model uses graph embeddings based on the ontology of the ATC and ICD-9 codes. The model initially pre-trains the embeddings on the single-visit data using self-supervised learning, similar to BERT (Devlin et al., 2019). The graph embeddings are learned using the Graph Attention technique (Velickovic et al., 2018), so as to learn hierarchical embeddings for each of the diagnostic and medication codes.

To test if we improve the performance on the original dataset, we evaluate overall accuracy metrics. For the medication recommendation task as shown in Table 3, in the MIMIC-III diagnostic code classification task through domain faithful deep learning parameter refinements, we *improve F1-score by 5.3%* with similar gains in Jaccard coefficient and PR-AUC and we *improve F1-score by 3.3%* on the medicine category classification task over the augmented dataset which contains counterfactual scenarios of in-category diagnostic codes, thereby increasing adherence to diagnostic-medication category mappings.

### 6.3 Box2D Car Racing

We also evaluate domain faithful deep learning on the Deep Q learning baseline for the Box2D environment in the CarRacing-v0 game (Brockman et al., 2016). This game is a classic car racing game, with the objective of keeping the car on the road, and the reward is -0.1 every frame and +1000/N for every track tile visited, where N is the total number of tiles visited in the track. For example, if you have finished 1 track tile in 732 frames, your reward is 1000 - 0.1*732 = 926.8 points. The episode finishes when all the tiles are visited. The car can also go outside the playfield - that is, far off the track, in which case it will get -100 reward and die.

**Baseline**: We use a Deep Q learning (DQN) (Mnih et al., 2013) baseline that crops the output pixel matrix to isolate the racing car and sensor readers. The 2d map contained in the state space with colors indicating road, green, and the car is flattened into a 1d array. We have 9 discrete possible actions: 7 steering (neutral,

3 right, 3 left) and 2 power, 50% brake or 33% gas. i.e. the car cannot steer and power on at the same time. The rules we enforce is to ensure that the car stays on the road, by augmenting data of negative samples for actions that result in the car going off the road, and the corresponding PID controller used in prior car navigation methods (Zhao et al., 2012; Nie et al., 2018). Further we optimize the same action-value function used in Q-learning over these augmented examples.

Table 4: Our method considerably increases the running average of the CarRacing-v0 reward consistently across the number of episodes of training.

| Episodes | DeepStochLog | Logic Distillation | DQN | DQN RA | DQN RA-WCR | DQN RA-WCR [dfdl] |
|---|---|---|---|---|---|---|
| 10 | $126.7 \pm 8.2$ | $154.2 \pm 7.6$ | $150.2 \pm 3.6$ | $161.3 \pm 4.8$ | $198.4 \pm 5.1$ | $\mathbf{225.7 \pm 5.5}$ |
| 50 | $182.6 \pm 5.4$ | $315.5 \pm 8.5$ | $310.8 \pm 4.7$ | $321.8 \pm 5.1$ | $335.5 \pm 6.4$ | $\mathbf{356.6 \pm 5.3}$ |
| 100 | $217.4 \pm 4.9$ | $329.7 \pm 9.7$ | $352.1 \pm 5.8$ | $361.7 \pm 6.0$ | $380.6 \pm 7.6$ | $\mathbf{396.3 \pm 6.4}$ |
| 200 | $238.3 \pm 6.1$ | $415.2 \pm 11.3$ | $672.5 \pm 6.9$ | $684.4 \pm 6.7$ | $696.3 \pm 8.9$ | $\mathbf{711.9 \pm 7.6}$ |
| 500 | $296.5 \pm 9.7$ | $510.8 \pm 13.7$ | $827.3 \pm 7.1$ | $831.3 \pm 7.4$ | $844.6 \pm 9.6$ | $\mathbf{850.3 \pm 9.9}$ |
| 750 | $410.6 \pm 12.3$ | $627.3 \pm 15.0$ | $831.5 \pm 7.4$ | $841.2 \pm 8.6$ | $851.9 \pm 10.1$ | $\mathbf{862.2 \pm 10.2}$ |

**Early rewards** The value of incorporating domain constraints of the road outline, similar to the variants explored in medical recommendations, can be seen in early episodes where DQN takes up to 100 episodes to achieve a reward of 350, whereas with DFDL, we achieve it in 50 episodes. Further, we see consistent increase through all episode checkpoints as shown in Table 4.

### 6.4 Toxicity Detection

In the domain of toxicity detection, the ability to identify hateful comments in online media is of importance to mitigate harms. To this end, prior work has focused on building text classifiers to label text as hate or not-hate. Since the ability of such text classifiers is limited by the coverage of the dataset it is trained on, they often misclassify new and emergent toxic sentences. These errors can be high-stakes in that they may be targeted at sensitive demographic groups, such as minors. In these scenarios, the domain practitioner might prefer to err on the side of classifying sentences in a conservative manner - i.e a higher weight on false negative errors rather than false positive (where positive class is hate). We experiment with a list of curse words as a way to introduce such domain specific priors, where we fine-tune a a baseline toxicity classifier (PerspectiveBERT) to be able to minimize such false negative errors when one of the curse words listed is present in the text (Vidgen et al., 2021). We see that while the classifier performs reasonably well in the traditional train/test splits of the data, it under-performs significantly worse on an adversarially created dataset where the curse words are inserted in sentences with high probability of being labeled as not hateful. In Table 6, we further see that the same model when trained with data augmentation and within-category-regularization, then the performance improves significantly. The categories in this domain are the five dimensions namely - toxicity, insult, threat, hate, and obscenity.

Table 5: Improvement in AUC of Toxicity detection task using domain faithful Perspective BERT models.

| Held-out safety rule $\rightarrow$ | Toxic | Obscene | Threat | Insult | Hate |
|---|---|---|---|---|---|
| Perspective-BERT | $0.72 \pm 0.03$ | $0.78 \pm 0.02$ | $0.71 \pm 0.06$ | $0.69 \pm 0.05$ | $0.73 \pm 0.03$ |
| DeepStochLog | $0.68 \pm 0.02$ | $0.71 \pm 0.05$ | $0.67 \pm 0.02$ | $0.68 \pm 0.04$ | $0.70 \pm 0.04$ |
| Logic distillation | $0.74 \pm 0.04$ | $0.78 \pm 0.03$ | $0.73 \pm 0.02$ | $0.72 \pm 0.01$ | $0.76 \pm 0.01$ |
| Perspective-BERT + RA | $0.80 \pm 0.03$ | $0.82 \pm 0.04$ | $0.76 \pm 0.02$ | $0.74 \pm 0.04$ | $0.75 \pm 0.01$ |
| Perspective-BERT + RA + WCR | $\mathbf{0.89 \pm 0.04}$ | $\mathbf{0.92 \pm 0.02}$ | $\mathbf{0.89 \pm 0.01}$ | $\mathbf{0.87 \pm 0.03}$ | $\mathbf{0.86 \pm 0.03}$ |
| Perspective-BERT + RA + WCR [dfdl] | $\mathbf{0.90 \pm 0.04}$ | $\mathbf{0.90 \pm 0.05}$ | $\mathbf{0.91 \pm 0.02}$ | $\mathbf{0.88 \pm 0.03}$ | $\mathbf{0.89 \pm 0.01}$ |

Table 6: Improvement in Accuracy of Toxicity detection using domain faithful Perspective BERT models.

| Held-out safety rule → | Toxic | Obscene | Threat | Insult | Hate |
|---|---|---|---|---|---|
| Perspective-BERT | $0.78 \pm 0.05$ | $0.81 \pm 0.02$ | $0.75 \pm 0.04$ | $0.73 \pm 0.04$ | $0.80 \pm 0.03$ |
| DeepStochLog | $0.72 \pm 0.01$ | $0.76 \pm 0.04$ | $0.71 \pm 0.03$ | $0.71 \pm 0.02$ | $0.73 \pm 0.02$ |
| Logic distillation | $0.80 \pm 0.04$ | $0.81 \pm 0.05$ | $0.77 \pm 0.07$ | $0.70 \pm 0.04$ | $0.77 \pm 0.01$ |
| Perspective-BERT + RA | $0.85 \pm 0.02$ | $0.86 \pm 0.04$ | $0.82 \pm 0.01$ | $0.80 \pm 0.04$ | $0.83 \pm 0.02$ |
| Perspective-BERT + RA + WCR | $\mathbf{0.91 \pm 0.02}$ | $\mathbf{0.94 \pm 0.04}$ | $\mathbf{0.94 \pm 0.02}$ | $\mathbf{0.92 \pm 0.03}$ | $\mathbf{0.91 \pm 0.04}$ |
| Perspective-BERT + RA + WCR [dfdl] | $\mathbf{0.94 \pm 0.02}$ | $\mathbf{0.96 \pm 0.02}$ | $\mathbf{0.96 \pm 0.01}$ | $\mathbf{0.95 \pm 0.03}$ | $\mathbf{0.97 \pm 0.01}$ |

# 7 Conclusion, Limitations, and Future Work

In this paper, we have outlined a domain-specific deep learning framework based on modular components of regularization and data augmentation. We leverage the benefits of this decoupling between functional specifications and batched execution afforded by our framework, to enable continuous refinement of parameters by domain experts, including optimizing hyperparameters of the constraints for overall accuracy and robustness in a synthetic simulation task, and 3 real world tasks - MIMIC-III medication recommendation, Car-Racing Box2D reinforcement learning, and toxicity detection. Here, we show that learning domain faithful deep learning models outperform models that do not incorporate domain knowledge, or those that do not optimize the hyper-parameters involved.

Our work extends on prior work of incorporating discrete predicates into deep learning models. While we build modular components involving data augmentation and regularization, other domain specific constraints that involve first order differential equations and converting them into regularization or other forms of physics-inspired modeling are left for future work. Additionally, if the domain specific constraint we enforce by definition is biased towards specific spurious correlations when augmented, our methodology might amplify and hurt generalization.

Also, we currently assume that we can alter one of the inputs that affects one of the rules. In the scenario where it ends up altering multiple rules, the perturbations are not optimized for in the data augmentation process, and the domain faithfulness might be violated. For these, we rely on the domain experts to carefully evaluate the constraint, and specify the behavior for multiple rules, before implementing in our method. Further, we have not explored how rule complexity affects the projection network design. Multi-hop or deep reasoning rules may require larger or more expressive architectures to capture compositional dependencies. Future work will study adaptive network designs that adjust capacity to the complexity and depth of the enforced rules.

## Broader Impacts

**Positive Societal Impact:** The proposed method has the potential to make machine learning systems more reliable and aligned with real-world domain expertise. In healthcare, it can reduce diagnostic errors by enforcing medically validated mappings. In content moderation, it improves robustness against adversarial inputs (e.g., disguised hate speech), supporting safer digital environments. More broadly, the method supports transparent collaboration between domain experts and ML systems, which may accelerate the adoption of AI in sectors that currently resist opaque models.

**Potential Negative Societal Impact:** There are several risks associated with the deployment of this framework.

- **Misuse of rules:** If domain rules are incorrect, biased, or maliciously crafted, they may encode harmful assumptions or exacerbate existing biases. This is particularly risky in sensitive domains such as healthcare and social platforms, where biased rules could result in health disparities or disproportionate censorship.

- **False sense of security:** The inclusion of rules might give an illusion of safety or correctness even when the model underperforms outside the rule-specified space.

- **Overreliance on expert heuristics:** In domains where domain knowledge is anecdotal or unvalidated, this framework may amplify unscientific or outdated practices.

- **Security risks:** Adversaries could reverse-engineer or exploit the rule structure to produce adversarial inputs that fool the model, especially in settings like toxicity detection.

**Mitigation Strategies:** We recommend the following safeguards for addressing these potential risks of negative social impact.

- **Auditing and validation:** Rules used for training should be rigorously validated, ideally through transparent documentation or peer review. Particularly in medical or social applications, fairness and demographic generalization audits should be conducted.

- **Human-in-the-loop supervision:** In high-stakes settings, rules and models should be used to assist rather than replace expert decision-making.

- **Rule provenance tracking:** All rules should be accompanied by metadata about their origin, coverage, and reliability, to discourage the use of unverifiable heuristics.

- **Limiting access:** The framework's flexibility should be gated in certain applications, such as surveillance or automated law enforcement, where misuse could lead to significant harm.

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

Table 7: Training error to measure overfit in each method on synthetic conditioned models

| Model Version | RMSE to ground truth |
|---|---|
| Baseline | 0.16 (0.14, 0.18) |
| Baseline+Mapped | 0.20 (0.17, 0.23) |
| Baseline+Predicate | 0.21 (0.18, 0.24) |
| DeepStochLog Winters et al. (2022) | 0.05 (0.03, 0.07) |
| Logic Distillation Hu et al. (2016) | 0.06 (0.05, 0.07) |
| RA | 0.08 (0.06, 0.10) |
| RA-WCR | 0.10 (0.08, 0.12) |
| RA-WCR [dfdl] | **0.09 (0.06, 0.12)** |

Table 8: Training accuracy of G-BERT models on the MIMIC-III medication recommendation task.

| | Model | Jaccard | F1 | PR-AUC |
|---|---|---|---|---|
| Original | G-Bert | 0.83 ±0.02 | 0.91 ±0.05 | 0.83 ±0.04 |
| | G-Bert+Cat | 0.81 ±0.04 | 0.89 ±0.02 | 0.81 ±0.03 |
| | G-Bert+Mapped | 0.80 ±0.03 | 0.89 ±0.04 | 0.79 ±0.04 |
| | G-Bert+Predicate | 0.80 ±0.02 | 0.87 ±0.02 | 0.79 ±0.03 |
| | G-Bert RA | 0.78 ±0.03 | 0.88 ±0.05 | 0.81 ±0.03 |
| | G-Bert RA-WCR | 0.77 ±0.03 | 0.88 ±0.04 | 0.80 ±0.03 |
| | G-Bert RA-WCR [dfdl] | **0.78** ±0.01 | **0.89** ±0.01 | **0.81** ±0.03 |

Table 9: Training-time CarRacing-v0 reward to measure overfit.

| Episodes | DeepStochLog | Logic Distillation | DQN | DQN RA | DQN RA-WCR | DQN RA-WCR [dfdl] |
|---|---|---|---|---|---|---|
| 10 | 218.3 ± 4.6 | 231.4 ± 3.8 | 273.5 ± 3.0 | 294.1 ± 3.7 | 291.5 ± 4.7 | **290.5 ± 3.1** |
| 50 | 254.3 ± 3.4 | 370.6 ± 4.7 | 350.1 ± 2.1 | 383.1 ± 2.8 | 381.7 ± 4.1 | **385.1 ± 3.8** |
| 100 | 285.0 ± 2.9 | 401.5 ± 5.3 | 426.9 ± 3.1 | 501.6 ± 4.1 | 492.4 ± 3.8 | **500.8 ± 2.1** |
| 200 | 412.7 ± 5.2 | 453.7 ± 7.1 | 701.2 ± 4.2 | 761.7 ± 5.8 | 753.6 ± 4.1 | **760.0 ± 2.5** |
| 500 | 645.9 ± 6.2 | 753.1 ± 8.2 | 833.4 ± 6.3 | 880.5 ± 4.7 | 840.4 ± 3.2 | **849.2 ± 3.7** |
| 750 | 847.7 ± 6.3 | 916.3 ± 4.7 | 961.7 ± 6.0 | 971.5 ± 4.1 | 943.8 ± 4.6 | **955.2 ± 4.8** |

