# OpenReview forum: "A Modular Abstraction for Integrating Domain Rules into Deep Learning Models"
_TMLR — Rejected by TMLR_

### Review · Reviewer_Q1Bh · 2025-09-08

**Summary Of Contributions:**

The paper tackles the important problem of incorporating domain knowledge into the training of deep learning models. The proposed mechanism to achieve this is by specifying rules over subgroups of data that are then used to automatically perform data augmentation and regularization.

The authors perform experiments on four domains (a synthetic task, a medical recommendation task, a RL environment, and toxicity detection), showing the generality of the framework. The authors find that the proposed framework allowed them to outperform the baselines.

Strengths:
- The problem tackled by the paper is important for many applications.
- The scope of the experiments shows the generality of the framework.

Weaknesses:
- Key aspects of the framework (e.g., scope and assumptions) are not described in sufficient level of detail (neither in the main text, nor in the appendix).
- The lack of a user study and concrete examples makes it difficult to estimate the amount of effort required by domain experts to leverage the framework.

**Audience:**

Yes

**Audience Explanation:**

I believe TLMR's audience would be interested because of the importance of the problem it tackles. Furthermore, the proposed solution appears to be broadly applicable.

**Broader Impact Concerns:**

I have no particular concerns that are not addressed in the Broader Impact Statement section.

**Claims And Evidence:**

No

**Claims Explanation:**

The claims in the paper are generally well supported with clear, convincing evidence, with a couple of significant exceptions.

**Unclear aspect 1:** the paper lacks a precise unifying mathematical or computational description of the rules that can be used in the framework.

The authors state

> it is essential to understand how to effectively specify such rules in a way that complements deep neural network (DNN) learning paradigms

Unfortunately, the manuscript in its current form does not support the claim that the framework solves this problem. The main reason, in my opinion, is that the manuscript does not describe a precise unifying mathematical or computational language in which the rules are specified by domain experts (or alternatively, the assumptions that the rules must satisfy).

Similarly, the Appendix contains a description of the rules used in the experiments, but in some cases the description is not detailed enough to deduce how the described rules map onto the framework. For example, one of the experiments is a RL task. How do Equation 1 and Algorithm 1 fit into the RL framework? Neither Algorithm 1 nor the data augmentation equation in section 3.2 seem general enough to accommodate a PID controller that generates data (e.g., a PID controller wouldn't map onto a uniform distribution over $C2I(c_x)$). Is this done over individual actions or over entire trajectories?

Also due in part to the lack of a precise unifying language to specify rules, the boundary between what the framework proposes and common practice is blurry. For instance, it is unclear if the paper is just a conceptual framework and every domain is implemented independently from scratch or if there is a unified implementation that is reused across all experiments.

**Unclear aspect 2**: the scope of the framework is never stated precisely.

For example, it is not immediately clear (and the manuscript does not discuss) how the RL formalism fits into the formalism in section 3, specifically the equation on page 4.

In a paper like this (which seems to place less emphasis on the framework as a computational system and more on the framework as a conceptual structure), I would expect a precise and well-motivated ontology of the class of problems that the framework tackles. However, the paper lacks a thorough discussion on the assumptions that a domain has to satisfy for the framework to be applicable.

**Unclear aspect 3**: The amount of effort and expertise required of the domain expert is not clear. E.g., it seems the domain expert has to understand what data augmentation is and how to perform it in the domain. A discussion on this would make it easier for the reader to understand the cost of using the proposed framework.

**Requested Changes:**

The paper would benefit a lot from:

1. A precise, unified mathematical or conceptual language to specify the constraints (otherwise, the paper should at least state the assumptions that the domain rules must satisfy, e.g., must they be differentiable?).

2. State which parts of the framework need to be specified by domain experts for each domain and which parts are general.

3. Discuss and state what classes of problems the framework is an adequate solution for (i.e., the scope of the framework), and (4) state what assumptions you are making from the domain expert (does the domain expert also have to be a deep learning expert?).

---

> ### Author Response · Authors · 2025-10-28
> **Response**
>
> We thank the reviewer for their feedback. We will address the issues highlighted in the revision, with additional results, and will do a thorough proof-read of the manuscript. Specifically on:
>
> > presentation in main paper (moved to appendix)
>
> We will improve the presentation of our key contributions, and assumptions in Section 3.
>
> > lack of user study
>
> We compare against state-of-the-art models used for each of the 3 real-world tasks evaluated, and measure the gains in accuracy and domain faithfulness. Beyond this, conducting a user study engaging with domain experts is unfortunately resource prohibitive for our project.
>
> > lack of precise mathematical or computational language
>
> We assume the existence of domain specific rules that correspond to a set of mappings in a shared input and output concept space in our problem formulation. (Table 1). Using this, we then integrate these rules into the training recipe of deep neural networks (Algorithm 1) by intervening at the batch sampling leveraging on-the-fly data augmentation based on the domain specific rules [dfdl]. We find that, without this critical contribution, simple data augmentation and regularization are not able to achieve the gains in Tables 2-6.
>
> > details on domain-specific implementations
>
> We will explain the individual instantiations of the Data Augmentation module in our framework in greater detail in the Appendix. Specifically, we embed our approach in the existing mechanisms in each of the tasks, rather than coming up with a new abstraction that domain experts need to learn. This way, we minimize the technical debt a domain practitioner needs to pay by reusing existing data generation pipelines in each of the 3 real-world tasks and repurpose them to fit into our framework.
>
> > boundary between framework and best practices
>
> The sampler that performs per-batch data augmentation shares a common abstraction that is implemented in JAX. The specific instantiations then extend from this abstract class by implementing domain-specific data generation sub-modules. We will provide more detail on this in the Appendix.
>
> > assumptions of the domain expertise
>
> We refer to mapping currently present in Table 1 and will add corresponding assumptions and implementation details of this framework that a domain has to satisfy to further demonstrate its generalizability.
>
> > amount of effort needed from the domain expert
>
> We will explain the assumptions on the implementation details for a domain to be integrated into our framework, such as, there exists an ontology of input and output concept classes, and a corresponding mapping between the two which are easily enumerable in a synthetic data augmentation pipeline.

---

### Review · Reviewer_BmiQ · 2025-09-16

**Summary Of Contributions:**

The authors of this work propose a framework to combine expert knowledge and deep learning. Specifically, the framework consists of a data augmentation combined with a penalized loss function that drive the deep learning model to a solution that match as closely as possible the constraint defined by the expert knowledge. Furthermore, the authors evaluate the performance of their model over one synthetic dataset and 3 real world datasets.

**Audience:**

Yes

**Audience Explanation:**

Combining expert knowledge and (observational) data to learn DL models is an extremely relevant in many context, especially in the healthcare field.

**Broader Impact Concerns:**

No concern

**Claims And Evidence:**

No

**Claims Explanation:**

The description of the experiments is too brief. It's essential to clearly and extensively explain the specific expert knowledge used in the real-world experiment, as this is the core focus of the article. Without this crucial information, readers can't interpret the results or understand why the model's performance improved.

**Requested Changes:**

1. Page 4 - Section 3.1: Why you defines the conditional probability as  P(c_x,c_y|c_x=c) instead of P(c_y|c_x=c)?
2. Page 5 - Section 3.2: Why you stated that you are using counterfactual data? The term counterfactual is used in the sense of causal inference described by Judea Pearl?
3. Page 8 - Section 5: You should add a citation or at least a footnote to the authors of JAX.
4. Tables 2,3,4: Please add the training set performance to show how much each model overfit the data.
5. Sections 5.2, 5.3, 5.4: Explain how you translated the expert knowledge in rules used by your framework.

---

> ### Author Response · Authors · 2025-10-28
> **Response**
>
> We thank the reviewer for their feedback. We will address the issues highlighted in the revision, with additional results, and will do a thorough proof-read of the manuscript. Specifically on:
>
> > Page 4, Sec 3.1: conditional probability
>
> We apologize for the typo, it is meant to be P(c_y|c_x=c), thus defining a conditional probability over the output concept class for a given input concept class.
>
> > Page 5 - Section 3.2: counterfactual data
>
> We use counterfactual in the sense of causal inference, but under a restrictive independence assumption that changes to a single input feature i.e, do operator, does not interfere with other input features, and measure the faithfulness of the model’s outputs to the domain-specific rules that are narrowly defined as mapping between individual input and output concept classes. Modeling interactions between a set of inputs and output is currently out of scope in our problem formulation. We will explain this in detail in Section 3.2.
>
> > Page 8 - Section 5: JAX citation
>
> we apologize for the omission, and include a citation
>
> > Tables 2,3,4: training overfit errors
>
> We will include the training set performance to demonstrate the overfit. Thank you for the suggestion.
>
> > Sections 5.2, 5.3, 5.4: translation of expert knowledge to our framework
>
> Table 1 provides a summary of how the different domain rules are translated into our framework. We will emphasize this in the text.

---

### Review · Reviewer_U4nq · 2025-10-14

**Summary Of Contributions:**

The paper proposes a domain-specific deep learning framework that integrates symbolic constraints via modular components for regularization and data augmentation. The authors claim that this decoupling allows domain experts to refine parameters and optimize constraint hyperparameters, leading to improved performance across synthetic and real-world tasks (MIMIC-III medication recommendation, Car-Racing Box2D reinforcement learning, and toxicity detection).
While the topic of neuro-symbolic AI is timely and relevant, and the reported results appear encouraging, the paper suffers from several conceptual, methodological, and presentation issues that significantly undermine its contribution.

**Strengths:**
- The topic of neuro-symbolic AI is highly relevant and aligns well with current research trends.
- The empirical results reported across multiple tasks seem promising.

**Weaknesses:**
- The proposed approach falls into the well-established category of symbolic knowledge injection (SKI), yet the paper lacks a proper overview of foundational work in this area (e.g., [1]).
- The novelty of the method is unclear. The data augmentation strategy appears to rely on simplistic sampling or noise perturbation, which is not convincingly differentiated from standard techniques.
- The theoretical section (Section 4) is poorly motivated and confusing, introducing adversarial training in the proof of Theorem 4.1 without prior discussion. It is unclear whether adversarial robustness is a core contribution of the paper.
- The experimental comparison omits several state-of-the-art NeSy models (e.g., DeepProbLog [5], Logic Tensor Networks [6], Relational NeSy Markov Models [7]).
- The related work section does not clearly position the proposed method against existing SKI and NeSy approaches (e.g., [2]).
- The paper contains numerous grammatical errors and typos, which detract from its readability and professionalism.

**Additional Comments:**

- The definition of the augmentation strategy in Algorithm 1 is vague and potentially misleading. If the authors are simply sampling other datapoints from the dataset that satisfy the same rule, this is not a novel augmentation technique.
- The assumption that perturbations preserve rule validity is questionable, especially in discrete domains like images.
- The distinction between functions g and h is unclear and should be clarified.
- The use of outdated baselines (e.g., G-BERT) weakens the empirical evaluation.


[1]. Ciatto, Giovanni, et al. "Symbolic knowledge extraction and injection with sub-symbolic predictors: A systematic literature review." ACM Computing Surveys 56.6 (2024): 1-35.

[2]. Magnini, Matteo, Giovanni Ciatto, and Andrea Omicini. "A view to a KILL: knowledge injection via lambda layer." WOA. 2022.

[3]. Lee, Jinhyuk, et al. "BioBERT: a pre-trained biomedical language representation model for biomedical text mining." Bioinformatics 36.4 (2020): 1234-1240.

[4]. Nentidis, Anastasios, et al. "Overview of BioASQ 2025: The thirteenth BioASQ challenge on large-scale biomedical semantic indexing and question answering." International Conference of the Cross-Language Evaluation Forum for European Languages. Cham: Springer Nature Switzerland, 2025.

[5]. Manhaeve, Robin, et al. "Deepproblog: Neural probabilistic logic programming." Advances in neural information processing systems 31 (2018).

[6]. Badreddine, Samy, et al. "Logic tensor networks." Artificial Intelligence 303 (2022): 103649.

[7]. De Smet, Lennert, et al. "Relational neurosymbolic Markov models." Proceedings of the AAAI Conference on Artificial Intelligence. Vol. 39. No. 15. 2025.

**Audience:**

Yes

**Audience Explanation:**

The general topic of neuro-symbolic integration and domain-informed learning is of interest to the TMLR audience. However, the current presentation and lack of novelty significantly limit the impact and relevance of the findings.

**Broader Impact Concerns:**

The authors have considered the broader societal impact of their work in a satisfactory manner.

**Claims And Evidence:**

No

**Claims Explanation:**

The evidence provided is not convincing due to the lack of clarity in the methodology, especially regarding the data augmentation strategy. The use of uniform sampling and random noise perturbation is not sufficiently justified or novel. Furthermore, the theoretical claims are not well-integrated with the rest of the paper, and the experimental comparisons are incomplete.

**Requested Changes:**

**Critical:**
- Provide a comprehensive overview of the SKI and NeSy literature, including foundational works such as [1].
- Clarify the novelty of the proposed data augmentation strategy. Explain how it differs from standard noise-based augmentation and why it is meaningful in the context of symbolic constraints.
- Improve the theoretical section (Section 4) by clearly stating its purpose, relevance, and connection to the rest of the paper. Avoid introducing concepts (e.g., adversarial training) without prior discussion. Specifically, clarify why adversarial training is introduced in the proof of Theorem 4.1, and whether adversarial robustness is a main contribution.
- Compare against more recent and relevant baselines in biomedical NLP (e.g., BioBERT [3], BioASQ [4]) and state-of-the-art NeSy models ([5], [6], [7]).
- Clearly distinguish the proposed method from existing SKI approaches such as [2].

**Recommended:**
- Thoroughly proofread the paper to correct grammatical errors and typos.
- Clarify ambiguous notations and definitions (e.g., the distinction between $g(M′(x))$ and $h(M′(x))$.
- Improve the presentation of corollaries and theoretical results to make their purpose and implications clear.

---

> ### Author Response · Authors · 2025-10-28
> **Response**
>
> We thank the reviewer for their feedback. We will address the issues highlighted in the revision, with additional results, and will do a thorough proof-read of the manuscript. Specifically on:
>
> > symbolic knowledge injection (SKI)
>
> We thank the reviewer for the suggestion, and we will compare the additional related work in this area.
>
> > novelty of the method
>
> Compared to existing data augmentation methods, we differ in 2 key areas:
> * Per-batch data sampling which increases the F1 scores and reduces RMSE as demonstrated in Tables 2-6. Our version [dfdl], when compared without this approach suffers due to batch variance that is unable to balance the tradeoffs of task accuracy and domain faithfulness (Eqn 1)
> * The data augmented relies on domain-specific concept mapping that operates over the concept space where the rules are mentioned.
>
> > theoretical section
>
> We apologize for the brevity in the proofs. We will introduce adversarial robustness first before using it. We leverage the results in the robustness domain to extend them to our restricted setting of discrete perturbations based on domain specific rules.
>
> > state-of-the-art NeSy baselines
>
> We thank the reviewer for identifying the missing baselines. However we note that DeepStochLog [1], included in our comparisons has been demonstrated to be superior to DeepProbLog, and Logic Tensor Networks, while achieving the same benefits of fuzzy logic in terms of computational complexity. Recent work on Relational neurosymbolic Markov models, model sequential decisions where the baseline models differ from the tasks we consider (including HMMs) and we find that it is orthogonal to our contribution. We will include a discussion of how it compares against additional SKI and NeSy approaches listed.
>
> > biomedical NLP baselines
>
> We thank the reviewer for these additional references. We find BioBERT, BioASQ to be similar in design to the baseline we used (G-BERT), but optimized to answer factoid questions as opposed to providing medication recommendations which are governed by ICD-9 and ATC ontologies. Without these taxonomies readily available in biomedical factoid datasets, enforcing domain-specific rules seems to not fit into our problem formulation.

---

### Decision · Action_Editor_JfT9 · 2025-12-05

**Recommendation:** Reject

**Audience:**

Yes

**Audience Explanation:**

Integrating domain-specific knowledge in deep learning is an important and timely topic for modern machine learning.

**Claims And Evidence:**

No

**Claims Explanation:**

The paper proposes a regularization strategy based on data augmentation over data subsets to integrate domain-specific knowledge into deep learning models. While the approach is interesting, all reviewers agree that the paper should more clearly articulate its relationships and differences with knowledge-integration methods from the neuro-symbolic literature. Moreover, the reviewers concur that the current claims are not sufficiently supported by experimental evidence, even after the rebuttal. In particular, the experimental analysis lacks a thorough comparison with existing neuro-symbolic approaches, as well as a discussion of the motivation and benefits of using data subsets and augmentation for knowledge integration as an alternative to other established strategies.

I agree with the reviewers that, although the proposed solution is promising, the paper requires substantial revision before it can be considered for publication. Therefore, I recommend rejecting the paper at this stage, with encouragement to resubmit after further improvements.

**Resubmission Of Major Revision:**

The authors may consider submitting a major revision at a later time.